# Distinguishing excess mutations and increased cell death based on variant allele frequencies

**Gergely Tibély**[1,2], **Dominik Schrempf**[2], **Imre Derényi**[2,3]*, **Gergely J. Szöllősi**[1,2,4]*

**1** MTA-ELTE "Momentum" Evolutionary Genomics Research Group, Budapest, Hungary, **2** Department of Biological Physics, Eötvös University, Budapest, Hungary, **3** MTA-ELTE Statistical and Biological Physics Research Group, Eötvös Loránd Research Network (ELKH), Budapest, Hungary, **4** Institute of Evolution, Centre for Ecological Research, Budapest, Hungary

* derenyi@elte.hu (ID); ssolo@elte.hu (GJS)

## Abstract

Tumors often harbor orders of magnitude more mutations than healthy tissues. The increased number of mutations may be due to an elevated mutation rate or frequent cell death and correspondingly rapid cell turnover, or a combination of the two. It is difficult to disentangle these two mechanisms based on widely available bulk sequencing data, where sequences from individual cells are intermixed and, thus, the cell lineage tree of the tumor cannot be resolved. Here we present a method that can simultaneously estimate the cell turnover rate and the rate of mutations from bulk sequencing data. Our method works by simulating tumor growth and finding the parameters with which the observed data can be reproduced with maximum likelihood. Applying this method to a real tumor sample, we find that both the mutation rate and the frequency of death may be high.

## Author summary

Tumors frequently harbor an elevated number of mutations, compared to healthy tissue. These extra mutations may be generated either by an increased mutation rate or the presence of cell death resulting in increased cellular turnover and additional cell divisions for tumor growth. Separating the effects of these two factors is a nontrivial problem. Here we present a method which can simultaneously estimate cell turnover rate and genomic mutation rate from bulk sequencing data. Our method is based on the estimation of the parameters of a generative model of tumor growth and mutations. Applying our method to a human hepatocellular carcinoma sample reveals an elevated per cell division mutation rate and high cell turnover.

## Introduction

Cancer is an evolutionary phenomenon within a host organism that unfolds on the timescale of years, but only becomes apparent in late stages and, as a result, is most often not directly

**Data Availability Statement:** All data is available in the manuscript and Supporting information files. Code is available at https://github.com/tg433/evolgenom.

**Funding:** GT and GJSz received funding from the European Research Council under the European Unions Horizon 2020 research and innovation programme under grant agreement no. 714774. ID and DS received no specific funding for this work. The funders had no role in study design, data collection and analysis, decision to publish, or preparation of the manuscript.

**Competing interests:** The authors have declared that no competing interests exist.

observed during large part of its evolution. Genomic sequencing data offers a window into the evolutionary processes underlying tumor development and progression. Analyzing widely available bulk sequencing data, where sequences from individual cells are intermixed, however, is challenging. Bulk sequencing can only essay mutation frequencies for a population of cells from each tumor sample and does not resolve the genotypes of individual cells, making even basic evolutionary parameters, like mutation rate and death rate, difficult to recover.

Consider the neutral model of tumor growth, where new mutations can appear with each cell division, while cells can also die for reasons such as lack of nutrients or immune reactions [1, 2]. This simple process can be described by two paramaters, the ratio of cell death and birth rates and the mutation rate per cell division. Despite the simplicity of such a model and a plethora of bulk tumor sequencing data, these parameters remain largely unknown, with estimates spanning several orders of magnitudes [2].

While tumor cells can contain a large number of mutations, it is not clear whether this is due to an elevated mutation rate or frequent cell death and birth. There are arguments for both cases [3–7], but distinguishing between these two alternatives is difficult because there are no known methods for direct measurements. As a result, estimating the mutation rate per cell division in human tumors must rely on assumptions about the duration of the cell cycle, the growth rate of the tumor and the total mutational burden of the tumor [8–10].

In previous works, Williams et al. [2] and Bozic et al. [3] showed that the combined effect of the mutation rate and the death rate can be estimated from the frequencies of neutral mutations that is readily available from bulk sequence data. Distinguishing excess mutations and increased cell death, however, still required external information or assumptions. In follow-up work, Williams et al. [11, 12] showed that separating these two quantities can be achieved by the bulk sequencing of multiple spatially disjunct samples from the same tumor, thus, resolving a coarse grained cell lineage tree. This approach, however, is inherently limited by the number of samples in its ability to resolve the cell lineage tree and, as a result, in its ability to distinguish excess mutations from increased cell death.

In general, tumor phylogenies represent the evolutionary history of its subclones and can be used to test different hypotheses about tumor evolution. However, the specific features of cancer data pose challenges to the direct application of classical phylogenetic models. In particular, bulk sequencing data contain an unknown number of novel cancer genomes, while classical phylogenetic approaches assume that taxa are known a priori [13]. Tree deconvolution methods, for instance, attempt to solve this problem by combining phylogenetic inference with a deconvolution step, in which clonal subpopulations from mixed genomic samples are separated prior to or concurrent with inferring phylogenetic relationships between those subpopulations [13–17].

Our approach also models mutation accumulation along a tumor phylogeny, a cell lineage tree arising from a birth-death process described by the death-to-birth ratio of cells (characteristic of cell turnover). It does not, however, attempt to infer a tree. Rather, it attempts to approximate the approach conceptualized by Felsenstein, wherein the likelihood of a genetic data set is assessed by considering all possible genealogical histories of the data, each in proportion to its probability [18, 19]. As a result, in contrast to methods such as tree deconvolution and other clone tree methods [15, 16] that attempt to infer complete or partial tumor phylogenies, no single tree is inferred. Instead, the parameters of an explicit probabilistic model of both mutations and the tree along which they occur are estimated by approximating the average over all possible trees.

Below we demonstrate, as a proof-of-concept, that approximating the average over all possible trees can differentiate between a wide range of death rates (even very close to the birth rate), and accurately estimate the mutation rates spanning a range of several orders of

magnitude, based on a single tumor-normal sample pair. After introducing the underlying probabilistic model and the parameter estimation procedure we assess its accuracy on simulated data and present results on empirical data.

## Model

We describe the evolution of tumor cells in terms of a cell lineage tree, i.e., the bifurcating tree traced out by cell divisions. As cells that have died cannot be observed we consider the tree spanned by surviving cells. The leaves of this tree correspond to extant cells and the internal nodes to cell divisions. To model the descent of sampled cells, we consider a birth-death process conditioned to have a fixed number of observed lineages (in our case sequenced cells) [20, 21]. This process is parameterized by the birth rate (which is identical to the cell division rate and is also referred to as the cell turnover rate) $b$, death rate $d$, and number of sampled cells $n$. We measure branch lengths in number cell divisions, i.e., birth events. Consequently, the birth rate $b$ can be considered as a scaling constant that sets the unit of time. Changing the unit of time—an arbitrary decision of ours—should not change the shape of the tree. Therefore, rescaling the rates should have no effect. Thus, the relevant parameter that determines the structure of the cell lineage tree is the death-to-birth ratio $\delta = d/b$. We only consider exponentially growing cell populations (i.e., $\delta < 1$), the growth rates of which in units of cell birth are $(b - d)/b = 1 - \delta$. Mutations occur at a rate $\mu$ per site per cell division. They are considered neutral and we neglect the probability that a site is hit by mutation more than once.

Throughout the paper, the following notation is used: Branches of the cell lineage tree are denoted by the index $k$, the length of branch $k$ is denoted by $l_k$ and $L = \sum_k l_k$ denotes the sum of all branch lengths in the tree.

When simulating data we first sample random cell lineage trees with continuous-time branch lengths (measured in units of one over the birth rate) from a conditioned birth-death process parameterized by the growth rate $1 - \delta$ and the number of surviving lineages $n$ using the point process approach described in [20]. Mutations are subsequently simulated along branches of this tree: For each branch $k$ the number of mutations is drawn independently from a Poisson distribution with parameter corresponding to the product of the branch length $l_k$ and the mutation rate $\mu$, and appears in $n \cdot f_k$ cells of the final population, where $f_k$ is the fraction of cells that descend from branch $k$ (cf. Fig 1A) in the final population.

The data available from bulk sequencing are the mutant and wild type read counts at each site. To illustrate how such data can be used to separate the effects of the mutation rate and cell death intensity, consider the variant allele frequency (VAF) spectrum, which can be obtained from read count data. As shown in (Fig 1A and 1B) mutation frequencies in the population reflect the branch length distribution of the tumor's cell lineage tree, the leaves of which correspond to the population of cells at the time the sample is taken, and its root is the most recent common ancestor of these cells. Mutation frequencies in the population, however, are not directly observable. What is observed is a random sample of mutant and wild-type read counts per site. The ratio of the observed mutant and wild-type read counts (also called the variant allele frequency) on average corresponds to the fraction of mutant cells in the population, and the total number of reads is determined by sequencing depth.

Changing the death-to-birth ratio modifies the shape of the cell lineage tree by changing the relative lengths of branches closer to and further away from the root and (Fig 1A), as a result, modifies mutation frequencies in the population (Fig 1B and 1C) and the observed variant allele frequencies (i.e. changes the shape of the variant allele frequency spectrum or VAF, Fig 1D). Changing the mutation rate, on the other hand, does not have any effect on the branch length distribution as it changes the expected number of mutations uniformly along the tree.

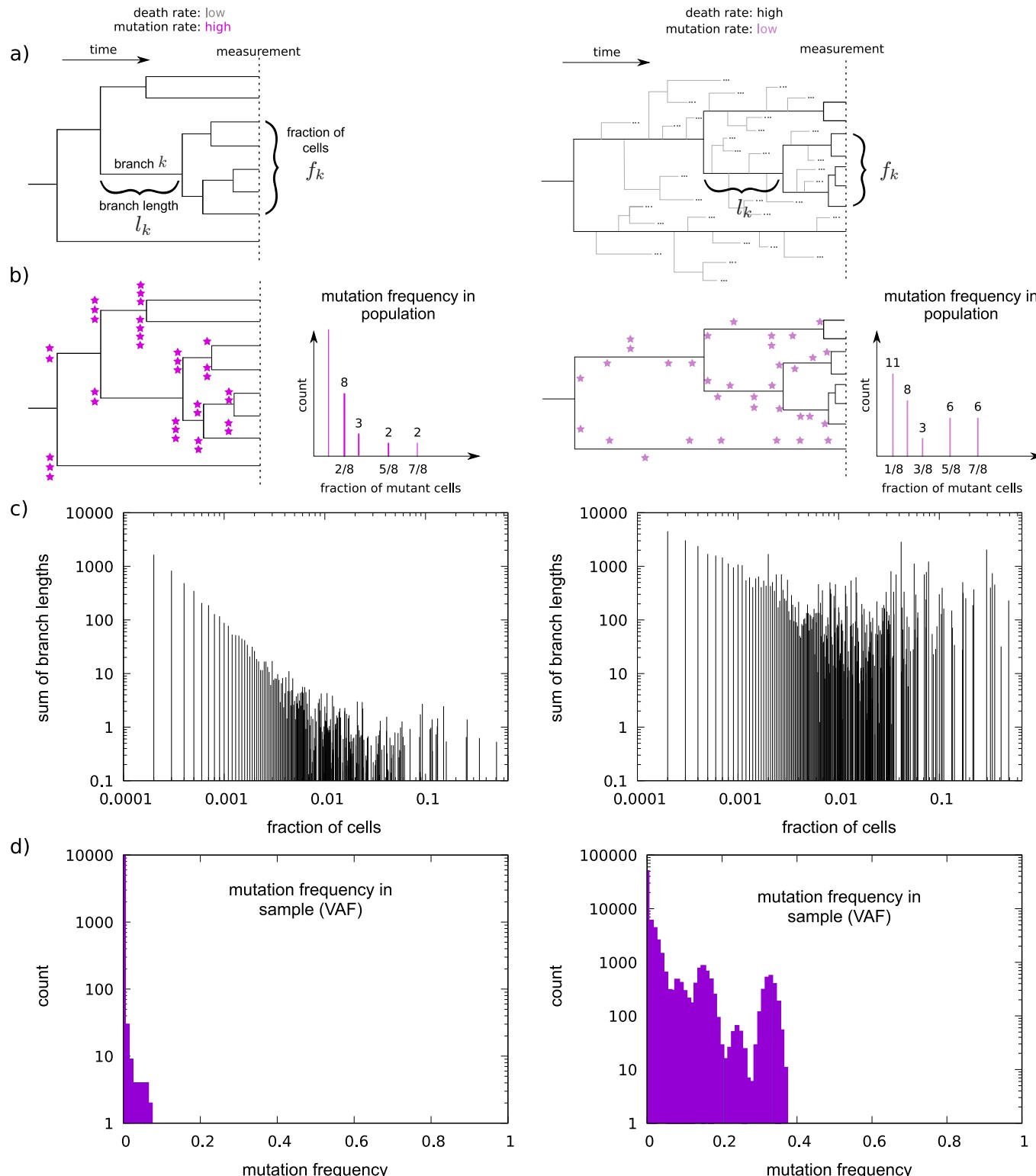

**Fig 1. Distinguishing mutation rate and cell death intensity based on variant allele frequencies.** Two possible scenarios for the generation of mutations along cell lineage trees. **A)** Different death-to-birth ratios lead to different lineage tree shapes. Bifurcations are cell divisions, leaves are cells comprising the bulk sequencing sample. Note that the (surviving) tree topologies are the same, only branch lengths differ. **B)** Mutations, symbolized by purple stars, accumulate at cell divisions. High death-to-birth ratio and low mutation rate can lead to the same number of observed mutations as low death-to-birth ratio and high mutation rate, however, the mutation spectrum of the two trees are different. **C)** For simulated trees of $10^4$ leaves, the differences in the branch length distribution are

clearly visible. **D)** VAF spectra for death-to-birth ratios of 0 (left panel) and 0.999999 (right panel). See also Fig A in S1 Text, which shows the effect of varying the mutation rate on the VAF spectra. The mutation rate was set to $\mu = 1$. Fractions of mutant cells are binned (note the logarithmic scale). Ploidy is set to two, contamination is zero. Simulated sequencing depth is 1000. The VAF spectra are based on the trees used to generate subplot **C)**.

Thus, changing the mutation rate leaves the relative frequencies of mutations in the population on average unchanged and, aside from the random sampling of reads, only changes the VAF by on average a multiplicative scaling factor (cf. Fig 1C and 1D and Fig A in S1 Text).

It should be noted, however, that sequencing data contains more information than the VAF spectrum. For each site the mutant and total read counts, and not only their ratio is known (e.g., 10 mutant reads out of 30 total reads have different statistical significance from 1 out of 3, even though they both correspond to a mutation frequency of 1/3).

To compare different combinations of mutation rates and death-to-birth ratios describing the observed empirical data we employ a likelihood based approach. First, we derive the likelihood $\mathcal{L}(D|\mu, \delta)$ of the observed data $D$, in our case the distribution of variant read counts, as a function of the per site mutation rate $\mu$ and the death-to-birth ratio $\delta$. As described below we maximize a score based on the likelihood function using samples of cell lineage trees drawn according to their probability corresponding to the conditioned birth-death process, in order to estimate the parameters that are most likely to have generated the observed data.

First we derive $\mathcal{L}(D|\mu, \mathcal{T})$ the likelihood of the observed data for a fixed cell lineage tree $\mathcal{T}$. Similar to other phylogenetic methods we assume that sites collect mutations independently of each other. This assumption is consistent with our assumption of neutral mutations, but is in general a simplification that is motivated by computational tractability. Consequently, $\mathcal{L}(D|\mu, \mathcal{T})$ takes the form

$$\mathcal{L}(D|\mu, \mathcal{T}) = \prod_i p(m_i|\mu, \mathcal{T}, r_i) \tag{1}$$

where the product runs over all sites, $m_i$ is the number of reads exhibiting a mutation at site $i$, and $r_i$ is the total number of reads covering site $i$. To calculate the probability of observing $m_i$ mutant reads out of a total of $r_i$ reads we consider the following two alternatives: First, if $m_i = 0$, then either a mutation occurred with probability $\mu \cdot l_k$ on some branch $k$ with length $l_k$, but no mutant read was observed out of the $r_i$ reads, or with probability $1 - \mu \cdot \sum_k l_k = 1 - \mu \cdot L$ no mutation occurred on any of the branches. Second, if $m_i > 0$, then a mutation occurred on some branch $k$ with length $l_k$ with probability $\mu \cdot l_k$ and $m_i$ mutant reads were observed out of $r_i$. Thus,

$$p(m_i|\mu, \mathcal{T}, r_i) = \begin{cases} \sum_k \mu \cdot l_k \cdot \mathrm{Binom}(0, r_i, f_k) + (1 - \mu \cdot L), & m_i = 0 \\ \\ \sum_k \mu \cdot l_k \cdot \mathrm{Binom}(m_i, r_i, f_k), & m_i > 0 \end{cases} \tag{2}$$

where $\mathrm{Binom}(m, r, f)$ is the binomial distribution of $m$ successes out of $r$ independent Bernoulli trials with success probability $f$, and $f_k$ denotes the fraction of sequenced leaf cells that descend from branch $k$. Multiple mutations at the same site are neglected, which is an appropriate approximation if $\mu \cdot L \ll 1$. In all of our applications we verified that this condition is satisfied.

To take into consideration sequencing errors, we must consider that they lead to an excess of spurious mutant reads that are in fact the result of sequencing error. We do this by introducing the probability $\varepsilon$ of a sequencing error per site per read. In the presence of sequencing error (i.e., $\varepsilon > 0$) a read can be i) an apparent mutant read caused by an actual mutation (a

true mutant read) or a sequencing error (a false mutant read), alternatively it can be ii) an apparent wild type read due to no mutation (a true wild type read) or because a mutation was reverted by a sequencing error (false wild type read):

$$
\begin{aligned}
p(m_i|\mu, \mathcal{T}, r_i, \varepsilon) \quad \approx \quad & \sum_k \mu \cdot l_k \cdot \mathrm{Binom}[m_i, r_i, f_k \cdot (1 - \varepsilon) + (1 - f_k) \cdot \varepsilon] \\
& + (1 - \mu \cdot L) \cdot \mathrm{Binom}(m_i, r_i, \varepsilon).
\end{aligned}
\tag{3}
$$

Note that this equation incorporates both the $m_i = 0$ and $m_i > 0$ cases.

So far, we have assumed that each site can have 2 states, wild type or mutant, corresponding to a DNA consisting of only two types of nucleotides, rather than four. It is, however, straightforward to introduce multiple mutant types. The likelihood function for 3 possible mutant types relevant for DNA sequences that we use for inference where indicated, can be found in the Materials and Methods section.

## Parameter inference

A conceptually straightforward approach for treating the unknown cell lineage tree as a nuisance parameter is to average over all trees $\mathcal{T}$ [18] according to their probability:

$$
\mathcal{L}(D|\mu, \delta) = \sum_\mathcal{T} \mathcal{L}(D|\mu, \mathcal{T}) \cdot p_{\mathrm{BD}}(\mathcal{T}|\delta),
\tag{4}
$$

where $p_{\mathrm{BD}}(\mathcal{T}|\delta)$ is the probability of the cell lineage tree $\mathcal{T}$ conditioned on a birth-death process with death-to-birth ratio $\delta$. Due to the very large number of possible trees the above average is intractable and because of the inherent lack of resolution resulting from bulk sequencing data a Markov Chain Monte Carlo sampler is impractical. To overcome these issues, here we resort to sampling a finite number of trees from the conditioned birth-death process with fixed $\delta$. However, as we can sample only a small fraction of trees, the estimated likelihood is typically dominated by a single tree, hence, the estimation becomes sensitive to the particular realization of the sample set. Using synthetic data (see below), however, we demonstrate that maximizing the average of the log-likelihood

$$
\ln \bar{\mathcal{L}}(D|\mu, \delta) = \frac{1}{|\mathcal{T}(\delta)|} \sum_{\mathcal{T}(\delta)} \ln \mathcal{L}(D|\mu, \mathcal{T}),
\tag{5}
$$

where $|\mathcal{T}(\delta)|$ is the number of sampled trees, allows accurate and robust inference that is less sensitive to sampling noise.

The mutation rate (defined as the number of mutations divided by the number of sites and the total branch length of the tree) is estimated directly from the data, in order to speed up the inference. Since some mutations are expected to be sequencing errors, we only count mutations of significant read count. The threshold number of mutant read counts is set in a sequencing depth dependent manner such that the expected number of sites with mutant reads that result from sequencing error is less than one for the entire dataset by choosing:

$$
m_{\mathrm{th}}(r, \varepsilon) = \min\{m : \mathrm{Binom}(m, r, \varepsilon) < \frac{1}{n_{\mathrm{sites}}}, m > \varepsilon \cdot r\},
\tag{6}
$$

where $r$ is the number of reads (i.e., the local sequencing depth) of which $m$ are mutant reads and $n_{\mathrm{sites}}$ is the total number of sites in the data. The expected total branch length of $\mathcal{T}$

corresponding to the significant mutant read counts can be estimated as

$$L_{\text{sig}}(\mathcal{T}, \varepsilon) = \sum_{r,k} \left[ \frac{n_r}{n_{\text{sites}}} \cdot l_k \sum_{m' \geq m_{\text{th}}(r,\varepsilon)} \text{Binom}(m', r, f_k) \right], \quad (7)$$

where $n_r$ is the number of sites with $r$ reads and $n_{\text{sites}}$ is the genome length (number of all the sites). Using the above formula, the mutation rate can be estimated as:

$$\mu_{\text{est}}(\delta, \varepsilon) = \mathbb{E} \left[ \frac{M_{\text{sig}}(\varepsilon)}{n_{\text{sites}} \cdot L_{\text{sig}}(\mathcal{T}, \varepsilon)} \right]_{\mathcal{T}(\delta)}, \quad (8)$$

where the $M_{sig}(\varepsilon) = \Sigma_{i, m_i \geq m_{\text{th}}(r_i, \varepsilon)} 1$ is the number of sites that have at least a threshold number of mutant reads in the data.

## Results on synthetic data

We validated the parameter estimation method on simulated data generated according to the model described above. The simulation procedure is described in the Materials and methods.

### No sequencing errors

Fig 2 shows how the estimated death-to-birth ratios and mutation rates compare to the true death-to-birth ratios and mutation rates. The method can reasonably differentiate between datasets with different death-to-birth ratios and mutation rates, and estimate their values.

Fig 3 shows the joint estimation of mutation rate and death-to-birth ratio pairs. The simulated data points are grouped into isolines with a constant number of observed mutations

$$M_{\text{obs mut}}(\mu, \delta) = \mathbb{E} \left[ \mu \cdot \sum_k l_k \cdot (1 - \text{Binom}(0, \bar{r}, f_k)) \right]_{\mathcal{T}(\delta)}, \quad (9)$$

where $\bar{r}$ denotes the average sequencing depth (average read count). The expected value is approximated by averaging over trees generated with death-to-birth ratio $\delta$.

Results on simulated data indicate that the accuracy of parameter estimates for different datasets are not uniform. As Fig F in S1 Text shows resolving large death-to-birth ratios requires trees with more leaves, and accuracy is increased when analyzing trees with a sufficiently large number of leaves. Aside of the effect of the size of the trees in case of high death-to-birth ratios, differences between estimates for different datasets can also result from: (i) sampling noise from the trees used in Eq (5), (ii) stochasticity of the simulated data and (iii) the simulated cell lineage tree used to generate the read counts. To differentiate between these possible factors, we chose a dataset with an estimated death-to-birth ratio that markedly deviated from the true value (dataset no. 7 for $1 - \delta = 1.0$ in Fig 2 with an estimated value of $1 - \delta = 0.47$). We calculated the estimated death-to-birth ratio values using 10 independent sets of 1000 trees. Estimates were obtained in the range of $1 - \delta \in [0.39, 0.54]$. Therefore, the deviation of the estimate from 1.0 cannot be attributed to the sample of fitting trees. Then, we simulated 10 additional datasets using the same tree as for the original dataset. The estimated death-to-birth ratio values were: $1 - \delta \in [0.44, 0.49]$, even more closely matching the original estimate. Consequently, the effect does not depend on the simulated mutations but on the tree along which they were generated. This suggests that the deviation of the estimates from the true death-to-birth ratios primarily reflects the fluctuation of the shapes of the trees used for sample generation. It seems that estimating the distribution of bifurcation times of the generating tree is not an easy task.

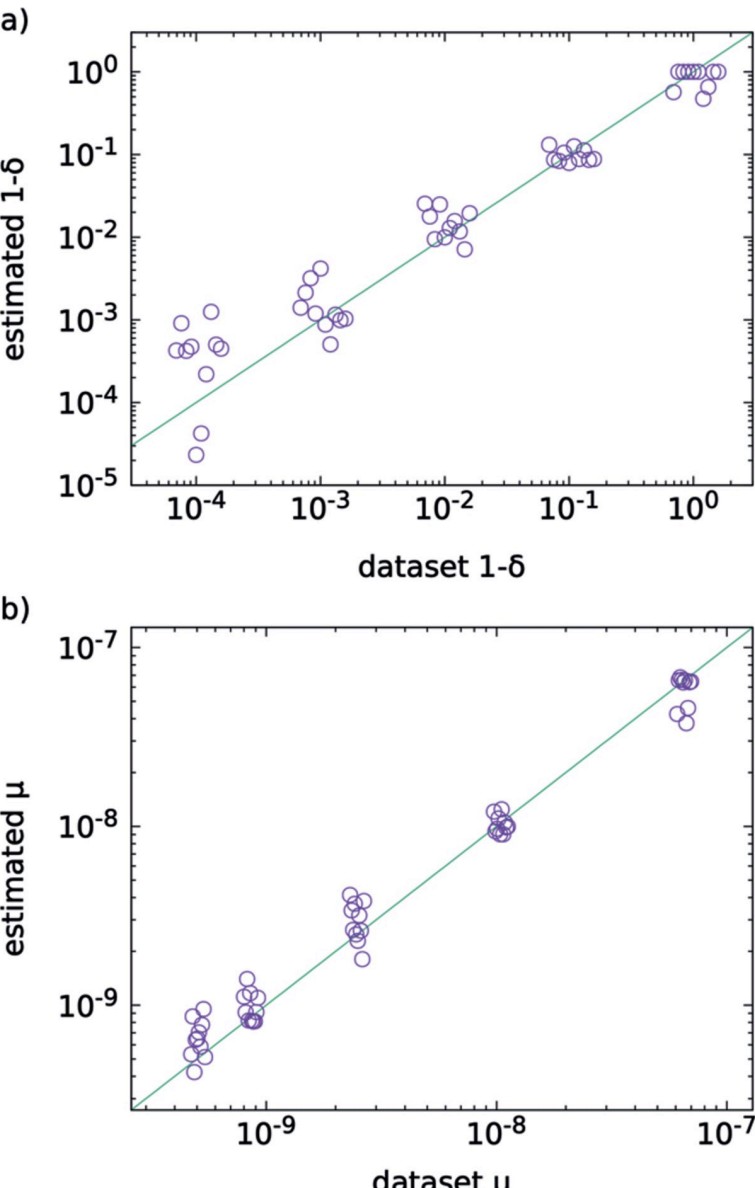

**Fig 2. True vs. estimated death-to-birth ratios and mutation rates.** In subplot A), the growth rate $1 - \delta$ (in units of the birth rate) is shown for better visualization. In both A) and B), 10 synthetic datasets were generated for each true value, with an independent cell lineage tree and associated mutations for each replicate. For each replicate $10^4$ trees with $10^4$ leaves were used for fitting (see the Materials and methods for details on calculating the likelihood). Points are slightly dispersed horizontally for clarity. Datasets are the same for A) and B). Horizontal ordering of the data points is the same for both subplots, e.g., the rightmost point in each group of points corresponds to dataset no. 10 in both plots.

## Effects of sequencing error

To estimate the effect of sequencing errors we re-estimated death-to-birth ratios while varying the magnitude of sequencing error, but using the same initial read counts as in Fig 2). In this case, the error rate of the data was also estimated by the fitting procedure, along with the mutation rate and the death-to-birth ratios. The range of the error rates is from $10^{-3}$, which is cited as the error rate of the Illumina sequencing technology [22], to $10^{-8}$, which is what advanced

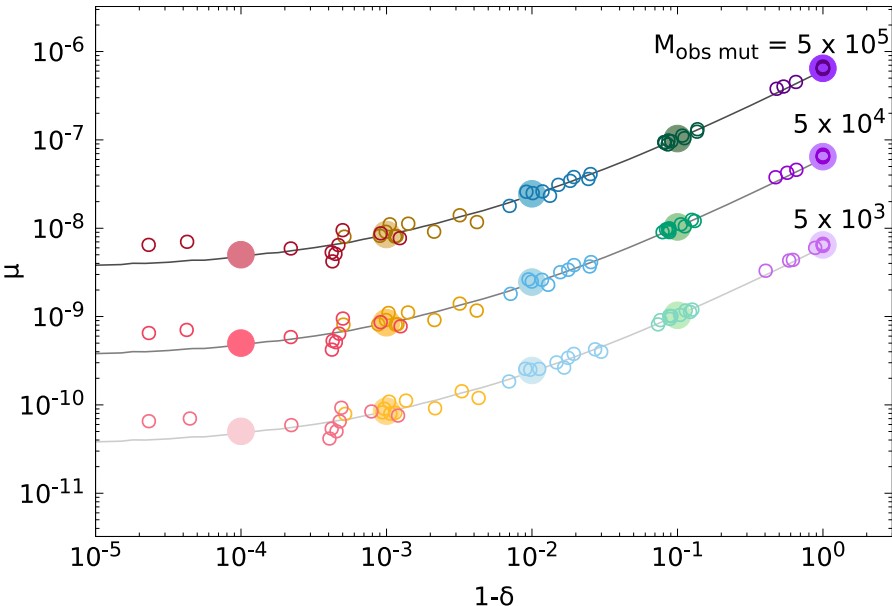

**Fig 3. Joint estimates of mutation rate and death-to-birth ratio pairs.** 10 synthetic dataset replicates were generated, each based on an independent cell lineage tree and associated mutations. For each true parameter pair replicates are denoted by the same color. True parameter values are indicated by large filled circles. Solid lines show the numerical approximation of parameter pairs for different values of $M_{obs\ mut}$. Data around the middle line are the same as in Fig 2. Note that the death rate decreases along the horizontal axis.

technology can achieve [23]. The influence of sequencing errors on the estimation of the death-to-birth ratios is shown in Fig 4. For an error rate of $10^{-3}$ (the case of the Illumina technology), the estimated death-to-birth ratios can have significant deviations from the true values.

## Results on empirical data

To estimate *in vivo* death-to-birth ratios and mutation rates, a tumor sample is required. Due to the sensitivity of our method to high sequencing error rates, we need a sample which is sequenced using a very low error rate technology. We found a partial whole genome sequnceing sample of a human hepatocellular carcinoma (HCC) [24], which was sequenced using the o2n sequencing technology [23], providing error rates between $10^{-5}$ and $10^{-8}$, which is significantly lower than the $10^{-3}$ rate of the standard Illumina process. Besides the low error rate, the dataset also has very high sequencing depth coverage, the average sequencing depth is 904, over 923383 sites. High sequencing depth allows the identification of more mutations and provides better resolved VAF spectra.

After preprocessing the data (see details in the Materials and methods), 2284 mutations were identified. The variant allele frequency spectrum is shown in Fig 5. Trees of $10^4$ leaves were used for estimating the parameters, as DNA was extracted and quantified from $10^4$ tumor cells that were precisely collected by laser capture microdissection (LCM) [24]. Using the procedure described above we obtained: a sequencing error of $\varepsilon = 10^{-7}$, consistent with the reported error rate of between $10^{-5}$ and $10^{-8}$ [23] for o2n, a death-to-birth ratio of $\delta = 1 - 1.1 \times 10^{-3}$, and a mutation rate of $\mu = 9.3 \times 10^{-8}$ per site per cell division. Fig 5 also shows the VAF of a synthetic sample, generated using the sample tree that fits best the empirical data.

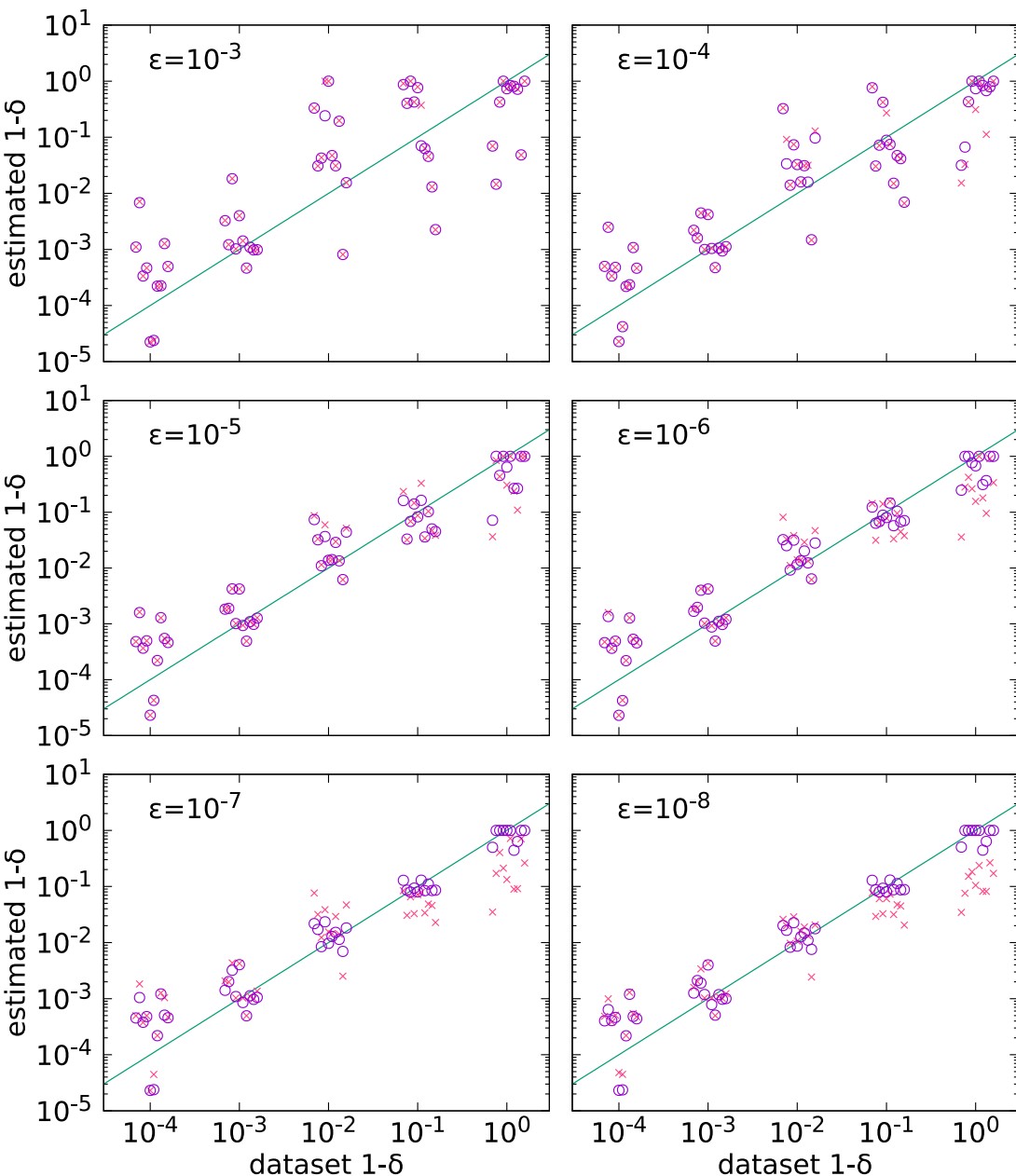

**Fig 4. The effect of varying the error rate.** Sequencing error rates are shown in the subplots. $10^4$ trees with $10^4$ leaves were used for fitting. Horizontal coordinates are slightly dispersed for clarity. Open circles are results corresponding to error rates fixed to their true values, crosses correspond to error rates estimated by the parameter fit. Each open circle-cross pair corresponding to the same dataset is vertically aligned. Note that the death rate decreases along the horizontal axis.

Fig C in S1 Text shows that the mutation rate and the death-to-birth ratio do not depend strongly on the value of the error rate.

Estimates for the mutation rate of healthy somatic cells can come from fitting mathematical models to the development of certain tumors, or cell line experiments, both methods observing mutations on a single gene [25]. A third method is to count the number of mutations in sequenced cells and estimate the number of corresponding cell divisions [26]. Finally, a

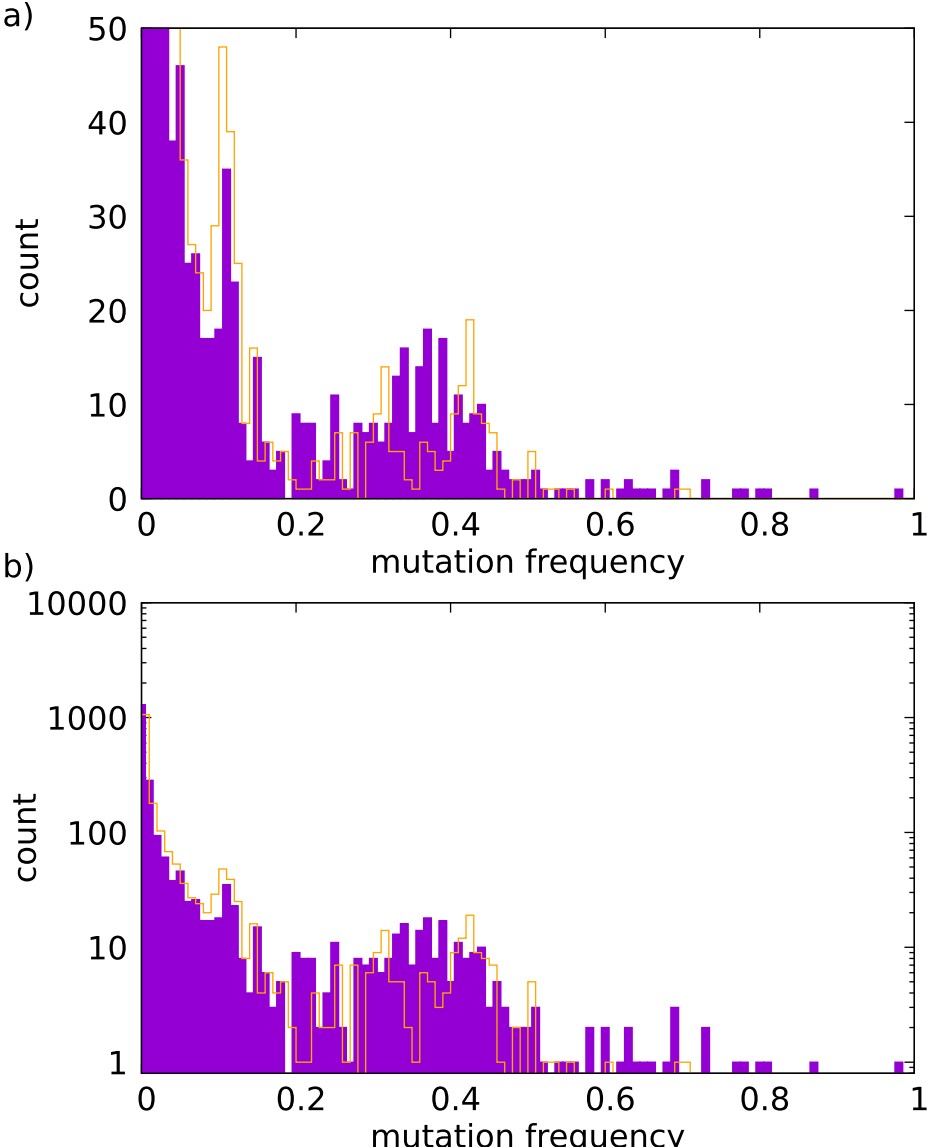

**Fig 5. Estimating the death-to-birth ratios and mutation rates on empirical data. A)** The solid purple bars show the VAF spectrum of a human hepatocellular carcinoma sample [24], obtained by o2n sequencing [23]. The orange line shows the VAF spectrum of a synthetic sample, generated using the tree having the highest likelihood for the empirical data. The tree has $10^4$ leaves corresponding to the number of sequenced tumor cells. **B)** counts are shown on a logarithmic scale.

mathematical model fitted to multiple samples from the same donor was applied to estimate mutation rates during early development in haematopoietic stem cells and neurons [11]. For all of these studies, estimated mutation rate values are in the range $10^{-9}$ to $10^{-10}$ per site per cell division [11, 25, 26], lower than our estimation of the hepatocellular carcinoma data.

Estimates of the mutation rate in tumors have high variance. In colorectal cancers, $\mu = 5 \times 10^{-10}$ per site per cell division [8] was estimated by observing the number of mutations and estimating the corresponding number of cell divisions. In kidney cancer, $\mu = 2 \times 10^{-9}$ per site per cell division was estimated [11], using multiple samples from the same tumor and fitting a mathematical model. The same study estimated $\mu = 5 \times 10^{-8}$ per site per cell division for lung

cancer. An effective mutation rate was estimated using a different mathematical model, in which the mutation and death rates are not separable [2]. Assuming no cell death ($\delta = 0$), for brain and prostate cancers $\mu = 9 \times 10^{-8}$ per site per cell division was estimated, along with $\mu = 8 \times 10^{-7}$ per site per cell division for lung and bladder cancers. Substituting our estimation of $\delta = 0.999$, these change to $\mu = 9 \times 10^{-11}$ and $8 \times 10^{-10}$ per site per cell division, respectively. Our estimation of $\mu = 9.3 \times 10^{-8}$ per site per cell division for HCC is at the higher end of the estimations for the mutation rates in tumors.

For the death-to-birth ratio, earlier estimations range within $\delta \in [0.08, 0.97]$ [11] for colon cancer, using the same methodology as above. In another study [3], $\delta = 0.72$ was estimated for fast-growing colorectal cancer metastases and $\delta = 0.999$ for premalignant colorectal tumors. Upper limits are provided [2] within $\delta \in [0.99, 0.999]$, assuming $\mu = 10^{-9}$ per site per cell division. Using our estimated mutation rate $\mu = 9.3 \times 10^{-8}$ per site per cell division, this changes to $\delta \in [0, 0.88]$. Our estimated death-to-birth ratio is similar to the previously estimated $\delta = 0.999$ of premalignant colorectal tumors. Possible causes for such an elevated death rate include the effect of the immune system, the deleterious nature of mutations, or competition for resources among tumor cells. In conclusion, for this tumor sample, the high number of mutations is due to a combination of an elevated mutation rate and a high death-to-birth ratio.

The above results allow us to estimate the number of cell division rounds from the founder cell to the biopsied tumor. The average height of the simulated trees with the estimated parameters is 2023 cell divisions. This value might seem counterintuitively low at first, according to the following estimation. A naive estimation of the average tree height would assume that to reach the size of $n_{\text{tumor cells}}$ leaves, the tree should have $\log_2 n_{\text{tumor cells}}$ branches between the root and a leaf. The average length of such a branch should be $1/(1 - \delta) = 1/(1.1 \times 10^{-3})$. As $n_{\text{tumor cells}} = 2.7 \times 10^9$ (see the next paragraph), the naive estimation of tree height is $\log_2(2.7 \times 10^9) \cdot (1/1.1 \times 10^{-3}) = 2.8 \times 10^4$. This reasoning, however, does not take into account the fact that trees with a death-to-birth ratio close to 1 have very different shapes compared to cell lineage trees resulting from a pure birth process. Consequently, the much lower tree height of 2023 as opposed to $2.8 \times 10^4$ can be attributed to the difference between the average tree with birth-to-death ratio $1 - 1.1 \times 10^{-3}$ and the average tree of a pure birth process stretched out by a factor of $1/(1.1 \times 10^{-3})$.

It is also possible to estimate the lifetime of the HCC sample and the cell division rate of the HCC tumor. The diameter of the tumor is 35 mm, while the length of a HCC cell is 25 $\mu$m [24]. This gives a total number of $2.7 \times 10^9$ cells in the entire tumor. The median HCC tumor volume doubling time is 86 days [27]. Based on these figures, the lifetime of the analyzed sample is around 7 years, and the cell division rate is estimated to be around $2023/[\log_2(2.7 \times 10^9) \cdot 86\text{days}] \approx 0.75 \text{ day}^{-1}$.

In the above calculation, the effect of sampling was neglected, which corresponds to cutting out exactly the descendants of one branch in the tumor lineage tree. This can be regarded as an approximation of taking a local sample from a solid tumor, and the estimated parameters refer to that branch of the lineage tree. Random sampling from a perfectly mixed population can be considered as an opposite extreme. In this case, the transformation of the birth rate is known [28]:

$$b = b'/\rho, \tag{10}$$

where $\rho$ is the sampling ratio and $b'$ is the birth rate corresponding to the $\rho = 1$ case. In our case, the size of the tumor is $2.7 \times 10^9$ cells, from which $10^4$ cell were sampled. Then the sampling ratio is $\rho = 10^4/(2.7 \times 10^9) = 3.7 \times 10^{-6}$ and the division rate is $b = 2.03 \times 10^5 \text{ day}^{-1}$, which is perhaps biologically less realistic.

## Conclusion

In summary, we describe a method to simultaneously estimate the mutation rate and the death-to-birth ratio together with the sequencing error rate, making it possible to answer the question which of the two is responsible for the elevated number of mutations in tumors. In particular, the mutated sites' read counts, which are closely related to the distribution of branch lengths and the shape of the cell lineage tree, contain useful information about the death-to-birth ratio, even in the presence of a moderate sequencing error rate.

Our results on simulated data show that the estimation method can resolve the death-to-birth ratio even if the birth and death rates are close to each other (i.e., $1 - \delta \ll 1$) as long as the sequencing error is sufficiently small. Unfortunately, for the error rates of standard Illumina sequencing, the estimation has a significant variance, therefore applying the method to typical samples is impractical. One solution is to use a sequencing technology with much lower error rates, e.g., as in Refs. [29, 30], or even below the $10^{-6}$ error rate of the PCR process [22, 23]. Another possibility is to apply noise filtering to standard sequencing data, e.g., deepSNV [31], and modify the error model of the fitting process accordingly.

As a proof of concept application we analyzed low error sequencing data from human hepatocellular carcinoma [23]. For both the death-to-birth ratio and the mutation rate we recovered estimates, a mutation rate being $9.3 \times 10^{-8}$ per site per cell division, and death-to-birth ratio $\delta = 0.9989$, which are higher than expected for most healthy tissues but fall within the range of previous estimates for tumors [2] and are consistent with a high mutation burden in HCC [32].

It is important to note that high death-to-birth ratios can produce marked subclonal peaks in the VAF spectrum (Fig 5). This implies that subclonal peaks are not necessarily the consequence of selection. Neutral processes, in particular a high death-to-birth ratio, can also produce such signal.

In this work, the death-to-birth ratio was assumed to be constant during the evolutionary process. It is more realistic to assume a death-to-birth ratio which changes during tumor growth [3]. In our case, the estimated strong cell death suggests that the tumor reached a slowly growing phase, in line with a Gompertzian model of tumor growth [33, 34], which is corroborated by the large sizes of observed tumors (diameter $\geq$ 1cm) used in the doubling time estimation [27]. It is possible that in earlier stages of tumor development, cell death is less frequent and doubling time is shorter. It might be the case that the rate of cell division is constant during tumor growth, and doubling time is set by the death-to-birth ratio, which, in turn, is limited by external factors.

The probabilistic model introduced here, together with the associated likelihood calculation and averaging procedure, including the simulation of cell lineage trees, can be extend in a relatively straightforward manner to consider the death-to-birth ratio as well as the mutation rate to be time or lineage dependent or both. Our current model has the potential to act as a null model compared to which models with increased complexity can be compared.

Lineage specific changes in the mutation rate or the death-to-birth ratio during the course of tumor evolution will both result in subtrees of the cell lineage tree that are described by different rates and descend from the cell in which the mutation rate or turnover rate changes. The two changes can be expected to have markedly different effects: a change in the mutation rate will from the perspective of the rest of the mutations observed act to "rescale" the branches of the subtree, while a change in the death-to-birth ratio will change the shape of the subtree together with the expected number of descendants surviving until the present. A decrease in the death-to-birth ratio (corresponding to an increase in $1 - \delta$, i.e., a positively selected "driver" mutation) is expected to have the most substantial effect on the mutations eventually

observed during tumor sequencing. As show in Fig G in S1 Text, despite a substantial lineage specific increase, the mutation rate is recovered relatively accurately. A systematic increase is, however, apparent in the inferred values of both the mutation rate and the death-to-birth ratio compared to "wild type" values resulting from the subtree with an increased death-to-birth ratio. This highlights the limitations [35] of the assumption of only neutral mutations, which our model shares with Williams et al. [2] and which is incompatible with recent empirical results on patterns of selection in cancer and somatic tissues [36].

Finnally, our model ignores any reminants of underlying tissue architecture and stem cell renewal mode that might impact the dyanmics of cell proloferation in a tumour. In heathly tissues it is clear that these effects play an imporant role, for instance, the organization of intestinal epithelial stem cells into crypts imposes constraints on clonal expansion resulting in elevated competition between stem cells belonging to the same crypt [37].

To go beyond a proof of concept application, aside of introducing "driver" mutations, several other important developments are required: At present we assume a simple uniform mutational process that is independent of genomic context and other mutational processes that shape the genomes of the clones comprising a tumor. In particular, taking into account mutational signatures [38] that incorporate a tumor's evolutionary context [17] based on the cell lineage tree offer the potential to increase both the realism and accuracy of our method. In addition, currently the method uses a well mixed population model for tumor growth and assumes uniform sampling. In the future, a more realistic growth model that includes spatial effects [39, 40] would enhance the applicability of the method opening up the possibility to use data from spatially resolved sampling of tissues, in which the measured mutation frequencies intertwine the correlated ancestry of sampled cells with the prevalence of the mutations.

## Materials and methods

### 3 mutant types

For 3 possible mutant types, relevant for DNA sequences, instead of the mutant read count $m$, we introduce three mutant read counts, corresponding to the three possible mutant types, $m^{(1)}, m^{(2)}, m^{(3)}$. Consequently, the input data need to contain three mutant read counts rather than one for each site. This leads to the use of a multinomial distribution, with four states: wild type and 3 mutant types. The possibility of more than one real mutation at the same site is still neglected, because it is very rare, technically a second-order process in the mutation probability of a site. We also neglect the probability of more than one error hitting the same site. The likelihood function at a single site is then

$$
\begin{aligned}
p(m_i^{(1)}, m_i^{(2)}, m_i^{(3)} | \mu, \mathcal{T}, r_i, \varepsilon) &\approx \\
&\approx \sum_k \mu \cdot l_k \cdot \frac{1}{3} \sum_{j=1}^{3} \text{Mult}\left( \left( m_i^{(j)}, m_i^{(j+1)}, m_i^{(j+2)}, r_i - \sum_{j'} m_i^{(j')} \right); \right. \\
&\quad \left. r_i; \; \left( p_\text{M}^{(j)}(f_k, \varepsilon), p_\text{M}^{(j+1)}(f_k, \varepsilon), p_\text{M}^{(j+2)}(f_k, \varepsilon), p_\text{w}(f_k, \varepsilon) \right) \right) + \\
&\quad + (1 - \mu \cdot L) \cdot \text{Mult}\left( \left( m_i^{(1)}, m_i^{(2)}, m_i^{(3)}, r_i - \sum_j m_i^{(j)} \right); \right. \\
&\quad \left. r_i; \; \left( p_\text{M}^{(j)}(0, \varepsilon), p_\text{M}^{(j+1)}(0, \varepsilon), p_\text{M}^{(j+2)}(0, \varepsilon), p_\text{W}(0, \varepsilon) \right) \right)
\end{aligned}
\tag{11}
$$

and

$$
\begin{aligned}
p_{\mathrm{M}}^{(j)}(f_k, \varepsilon) &= f_k \cdot (1 - \varepsilon) + (1 - f_k) \cdot \varepsilon/3 \\
p_{\mathrm{M}}^{(j+1)}(f_k, \varepsilon) &= f_k \cdot \varepsilon/3 + (1 - f_k) \cdot \varepsilon/3 = \varepsilon/3 \\
p_{\mathrm{M}}^{(j+2)}(f_k, \varepsilon) &= f_k \cdot \varepsilon/3 + (1 - f_k) \cdot \varepsilon/3 = \varepsilon/3 \\
p_{\mathrm{W}}(f_k, \varepsilon) &= f_k \cdot \varepsilon/3 + (1 - f_k) \cdot (1 - \varepsilon)
\end{aligned}
\tag{12}
$$

where $(j)$, $(j + 1)$, and $(j + 2)$ denote the three possible mutant types with cyclic notation $(j) = (j + 3)$, and Mult(vector of numbers of outcomes; number of trials; vector of probabilities of outcomes) is the multinomial distribution. The factor of 1/3 is due to the fact the if a mutation occurs at a site than the probability of each of the three possibilities is 1/3. In the multinomial distribution the first outcome corresponds to the true mutant type, the second and third to the false mutant types, and the forth to the wild type.

## Generating trees

Cell lineage trees are simulated by our own implementation [41] of the point process described in [20], which is more stable for death-to-birth ratios close to unity then the widely used Tree-Sim implementation [21].

## Generating synthetic samples

Simulated data consists of mutant and wild type read counts based on mutations generated along a cell lineage tree $\mathcal{T}$ obtained using the ELynx suite.

Mutations are generated as follows. For each site, each branch $k$ is checked for contributing a mutation with probability $1 - e^{-\mu l_k} \approx \mu l_k$. The first branch providing a hit is selected.

The total number of reads of the current site is drawn from a pre-specified distribution.

The number of mutant reads is drawn form a hypergeometric distribution, corresponding to the branch frequency $f_k$. Sequencing errors, if required, are introduced using a multinomial distribution, with probabilities $\varepsilon/3$, $\varepsilon/3$, $\varepsilon/3$, $1 - \varepsilon$ corresponding to the 3 possible false types and the true type.

The mutation rate for different death-to-birth ratios is chosen such that the total number of observed real mutations should remain close to each other, i.e., the estimation algorithm should have a similar amount of input data.

Our implementation of the synthetic data generation method is available at https://github.com/tg433/evolgenom.

## Calculating the likelihood

We calculate the average of the log-likelihood, Eq (5), on a fixed grid of death-to-birth ratios, where for each point we generate a sample of a given set of simulated trees using the ELynx suite [41]. At each grid point the error rate is optimized either using Brent's method [42] implemented in Julia's Optim package [43] or calculating the log-likelihood on a grid of error rate values to speed up the calculation. The mutation rate is estimated in both cases according to Eq (8). The final death-to-birth ratio is then estimated by interpolation using cubic splines and the final estimate for the mutation rate is based on this value.

Our implementation of the parameter estimation method is available at https://github.com/tg433/evolgenom.

## Preprocessing of the empirical data

The raw sequencing data was preprocessed according to [23], using the code provided by the authors. The DNA contents of $10^4$ cells were sequenced [24], along with a sample of neighboring normal tissue. Mutations were called using VarScan 2 [44], which is flexible and easy to adapt to the requirements of the fitting procedure. The original study [23] targeted 410k sites, however, the raw data covers well over $10^8$ sites. We applied the mapping quality and base quality thresholds of [23] to the data, and also left the minimum sequencing depth requirement of VarScan 2 at its default setting (8), both in the tumor and the normal samples. 923383 sites remained. Among these sites, the distribution of sequencing depths is wide, ranging from 8 to 10853, with a mean of 904. For mutation calling by VarScan 2, the minimum number of mutant reads was set to 1 and the strand filter switched off. Although the number of false positives increases with these parameter choices, the resulting called mutations correspond better to the error model of our fitting procedure than an error rate which changes sharply with threshold frequency or read count values. The minimum variant frequency was set to $10^{-6}$ to include even the least frequent mutation. Purity was set to 0.85, in accordance with [24]. We also checked that the default somatic p-value threshold does not exclude any candidate somatic mutations. Other parameters were left at their default values. Mutation frequencies were corrected for *i)* copy number variation (CNV), using VarScan 2 with default parameters, and *ii)* for ploidy of the sex chromosomes. CNV detection for targeted sequencing data is a more difficult task than for whole genome data, and VarScan 2 was found to be a stable performer [45]. We followed the CNV detection steps suggested by the VarScan manual. We used VarScan to obtain raw copy numbers, using the *copynumber* command. The tumor-to-normal ratio was set to 1.091 for this run. Adjustment for GC content and filtering for minimum region size was done using VarScan's *copycaller*, with default parameters. Finally, copy number data were smoothed and segmented using the *DNAcopy* circular binary segmentation algorithm in R. Sites having multiple variant types (i.e., number of reads of wild type plus most frequent mutant type being lower than the sequencing depth) were checked manually. Read counts of all 4 possible genotypes were identified for all variant sites.

## Supporting information

**S1 Text. Supplementary information for distinguishing excess mutations and increased cell death based on variant allele frequencies. Fig A: VAF spectra for varying mutation rates and death-to-birth ratios**. Branch length distributions for A) low death rate and B) high death rate and corresponding VAFs, respectively C) and D), with the mutation rate varying over two orders of magnitude. **Fig B: Loglikelihood-error rate curve of the empirical data**. The right panel shows the peak with a narrow loglikelihood range. **Fig C: Death-to-birth ratio-error rate and mutation rate-error rate curves of the empirical data**. The loglikelihood curve is shown in light gray. **Fig D: Loglikelihood-death-to-birth ratio and loglikelihood-mutation rate curves of the HCC data**. Interpolation between data points is by cubic splines. Green line highlights the maximum. **Fig E: The effect of tree sizes on the estimations**. Estimated death-to-birth ratios for fitted trees with 100 (top left), 1000 (top right), 10000 (bottom left), and 100000 (bottom right) leaves. Sizes of the sample generating trees are the same as those of the fitting trees. **Fig F: The effect of the error rate on the estimated mutation rate**. Sequencing error rates are $\varepsilon = 10^{-3}$ (top left), $10^{-4}$ (top right), $10^{-5}$ (middle left), $10^{-6}$ (middle right), $10^{-7}$ (bottom left), $10^{-8}$ (bottom right). $10^4$ trees with $10^4$ leaves were used for fitting. Horizontal coordinates are slightly dispersed for clarity. Open circles are results corresponding to error rates fixed to their true values, crosses correspond to error rates estimated by the parameter fit. Each open circle-cross pair corresponding to the same dataset is vertically

aligned. **Fig G: The effect of the error rate on the estimated mutation rate**. In 500 replicate experiments we introduced a "driver" mutation that increases $1 - \delta$ by 30%. The large purple point indicates the parameter values used to simulate the data, gray circles are inferences for data without driver mutations from [Fig 3](), and purple datapoints are inferences for data including drivers. A small, but systematic increase is apparent in the inferred values of both the mutation rate and the death-to-birth ratio compared to "wild type" values resulting from the subtree with an increased death-to-birth ratio driver lineage.
(PDF)

## Author Contributions

**Conceptualization:** Gergely J. Szöllősi.

**Formal analysis:** Gergely Tibély, Imre Derényi, Gergely J. Szöllősi.

**Funding acquisition:** Imre Derényi, Gergely J. Szöllősi.

**Investigation:** Gergely Tibély.

**Methodology:** Gergely Tibély, Imre Derényi.

**Software:** Gergely Tibély, Dominik Schrempf.

**Supervision:** Gergely J. Szöllősi.

**Validation:** Gergely J. Szöllősi.

**Visualization:** Gergely Tibély.

**Writing – original draft:** Gergely Tibély.

**Writing – review & editing:** Dominik Schrempf, Imre Derényi, Gergely J. Szöllősi.

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
