## [Decision Letter · Decision Letter 0]

8 Dec 2021

Dear Tibély,

Thank you very much for submitting your manuscript "Distinguishing excess mutations and increased cell death based on variant allele frequencies" for consideration at PLOS Computational Biology.

As with all papers reviewed by the journal, your manuscript was reviewed by members of the editorial board and by several independent reviewers. In light of the reviews (below this email), we would like to invite the resubmission of a significantly-revised version that takes into account the reviewers' comments.

We cannot make any decision about publication until we have seen the revised manuscript and your response to the reviewers' comments. Your revised manuscript is also likely to be sent to reviewers for further evaluation.

Sincerely,

Teresa M. Przytycka

Associate Editor

PLOS Computational Biology

Douglas Lauffenburger

Deputy Editor

PLOS Computational Biology

Reviewer's Responses to Questions

**Comments to the Authors:**

Reviewer #1: This paper seeks to disentangle the processes of new mutations due to a higher mutation rate or due to higher cell turnover rate. This work assumes a fixed mutation rate and a fixed cell turnover rate during the progression of a tumor, and seeks to estimate these two rates given variant and total read counts of mutations in bulk DNA sequencing data. The authors validate their method on simulated data, finding that while their method is sensitive to per-base sequencing errors, in regimes of low sequencing error the method provides accurate estimates of ground-truth mutation and cell turnover rates. Finally, the paper considers an HCC sample sequenced using a technology with low error, and discusses the estimated rates in light of related findings. While I find the work interesting, I have several comments and questions.

1. How do cell lineage trees relate to trees inferred using tree deconvolution methods?

Many clone tree inference methods have been proposed for reconstructing clone trees from bulk DNA sequencing data of tumors. See for example [1,2]. How do these trees differ from the cell lineage trees that you sample in your method? Is there a way to make use of these clone trees in your method?

2. Copy-number aberrations

VAFs are affected by copy-number aberrations (CNAs). I suggest using cancer cell fractions instead. Also, I would appreciate more details on how VAFs were corrected for CNAs for the HCC data.

3. Justify/discuss assumptions

There are two key assumptions: (1) fixed mutation rate and (2) fixed turnover rate. While the latter assumption is discussed in the Discussion, I would have appreciated simulation experiments that assess how your method would perform if this assumption was violated? Would you still be able to accurately infer the mutation rate?

Moreover, I would like to see a discussion on the assumption of a fixed mutation rate. It has been shown that the mutation rate can change during the evolution of a tumor, due to for instance mutations in DNA mismatch repair mechanisms. Or more generally, due to exposures to mutational signatures. For an example of the former, please see Ref [3].

4. More discussion on subclonal mutations clusters.

I found this point extremely interesting. I would appreciate more simulation experiments to further investigate this point. Also, please discuss this in light of the findings in the MOBSTER paper [4].

5. Extension to support multiple samples

It would be good if the method could support multiple bulk samples from the same tumor.

6. Software and documentation

- Please include example input files.

- Add comments to your code/functions.

- It looks like your code takes trees as input rather than generating them. Please describe how trees can be generated.

- To facilitate reproducibility of the experimental results, please include simulated data and preferably real data as well.

Minor comments:

- turnover or turn over?

- page 3: it'd be good to justify the assumption of independence among sites.

- Good to provide the name of the method in the paper.

- What is the correspondence between the number of mutations and number of leaves in the trees that you sample? More generally, how do you choose the number of leaves?

References

[1] El-Kebir M, Oesper L, Acheson-Field H, Raphael BJ. Reconstruction of clonal trees and tumor composition from multi-sample sequencing data. Bioinformatics. 2015;31(12):i62-i70. doi:10.1093/bioinformatics/btv261

[2] Deshwar AG, Vembu S, Yung CK, Jang GH, Stein L, Morris Q. PhyloWGS: Reconstructing subclonal composition and evolution from whole-genome sequencing of tumors. Genome Biol. 2015;16(1):35. doi:10.1186/s13059-015-0602-8

[3] Christensen S, Leiserson MDM, El-Kebir M. PhySigs: Phylogenetic Inference of Mutational Signature Dynamics. In: Biocomputing 2020. WORLD SCIENTIFIC; 2019:226-237. doi:10.1142/9789811215636_0021

[4] Caravagna G, Heide T, Williams MJ, et al. Subclonal reconstruction of tumors by using machine learning and population genetics. Nat Genet. 2020;52(9):898-907. doi:10.1038/s41588-020-0675-5

Reviewer #2: In this manuscript Tibély et al. propose a likelihood-based approach to estimate the mutation rate and cell renewal rate history of a tumor from DNA sequencing data from a single bulk sample of the tumor and an adjacent normal sample. The methodology used is sounding, even though the applicability and interest for a wide biological community is questionable and possibly limited due to the strong theoretical character and the various (unavoidable) strong assumptions implied. The method is applied to an experimental dataset, but there is a lack of tools for cross-validation of the inferred parameters with real estimates (even if the method is appropriate for synthetic datasets generated under the same methodological assumptions, which is not striking per se). The mathematical approach is clean but the text suffers from a too technical style that sometimes is more proper of a Technical Note than a Main manuscript. Altogether, in my view the authors should pay most attention on the narrative to better introduce the background and succinctly explain the model’s novelty on such a non-new question to be able to convince on its far-reaching biological implications.

MAIN POINTS

- The Introduction feels short and too simplistic given the level of concretion of the subsequent approach and it should be more pedagogical to acknowledge the multiple facets and limitations involved in the inference of evolutionary parameters. First, Williams et al. work is referred, but since those authors addressed a similar problem in their original publication and even refined their subsequent analysis based on multiple-sampling data, it is unclear what it was inappropriate in their methodology, or, in other words, what is genuine in this work’s approach that makes it that more successful at discerning mutation rate and growth-death ratio even when considering just from bulk sequencing data. Is it just the likelihood approach vs. machine learning methods? Is it the fact of simulating and fitting the whole VAF distribution rather than its global scaling properties derived from Luria-Delbrück model? Secondly, the method shows important limitations (and this is common to Williams et al. work): it assumes a net exponential tumor growth, which is not necessarily realistic, and purely neutral subclonal dynamics. Other authors have shown the prevalence of positive Darwinian selection across tumor types (Martincorena et al, 2017, Cell), and even if this is not incompatible with an excess of neutral passenger mutations, it would affect the subclonal dynamics and the tree shape, consequently affecting inferences from the VAF distribution, as has been largely debated (e.g. Tarabichi et al. 2018 Nat. Genet, as a criticism to Williams et al. 2016). These limitations cannot be ignored and have to be highlighted or authors may want to consider how much inferences would depart from the given estimated values when these assumptions get relaxed. One could draw the impression that estimates can result orders of magnitude deviated if basic evolutionary assumptions are inappropriate. For the same reason, I would avoid overclaims in the Discussion.

- The model definition is not sufficiently clear or precise. Authors use a lineage tree representation but should explain what methodology they use to generate the simulated trees, i.e. T(delta). Is it by Markov-chain Monte-Carlo? Continuous or discrete time? What is the initialization condition and end time? What are the random events? From the Methods it seems clear that tree topologies and mutagenesis are simulated independent from one another but a higher level of concretion in the main text would be helpful. In particular, there are two major confusing points:

- (i) Whether mutagenesis events are circumscribed to the cell division instant (i.e. internal nodes) or are simulated too all along branches. The text sometimes gives the impression of the former assumption, but at the same time the division rate is taken as unit of time. This is relevant insofar as in one case both parameters are intermingled and mu changes as a result of changes in delta, while in the other they represent independent measures (delta would impact the mutation burden just indirectly, by modulating the total no. of tumor branches, but would not affect its density) and thus parameters might be more easily resolvable? If mu is truly independent it should not depend on b.

- (ii) Related with the previous, how empirical VAF distributions (that are indicative of tumor subclonal composition) can be directly contrasted with a tree whose branches are said to represent individual cells rather than subclones of multiple cells. I would assume a representation in terms of a subclonal tree where branches represent populations of clonal cells would be more readily contrastable with experimental VAF data. It is unclear how the fitting procedure is done if branches are restricted to individual cells and not clades. As a suggestion, I think it would help to improve the explanation on Fig. 1. This shows site frequency spectra but these are difficult to relate with trees on the left as there is no sketch drawing read sites along the branches. And the paragraph describing this connection between site frequency spectrum and tree topology is confusing. On the same line, Fig. 1 would benefit much if it shows trees generated under four different scenarios (low mu-low delta, high mu-low delta, low mu-high delta, and high mu-high delta) or at least under scenarios where just one or the other parameter is changed but not both at the same time like is done here. That would better illustrate how the tree shape and mutation burden change with the two parameters of interest. Altogether, the model section is vague and contrasts with that of the parameter estimation procedure which is much more detailed (Indeed, the parameter estimation section is already initiated by the second last paragraph of pg. 3 even if the title comes later).

MINOR POINTS

- I in part differ from the diagnosis that the main limitation for evolutionary inference is that bulk sequencing does not resolve the genotypes of individual cells. I assume it is not as much a problem of spatial resolution as it is of temporal resolution, as it is limited to a snapshot. I think exploring models accounting for sampling bulk data at various serial time points is a suitable direction for future work. This is poorly explored. On the contrary, I am not persuaded of what the benefit of sampling individual cell genotypes is vs. sampling subclones in regard to tree reconstruction and evolutionary inferences. This is perhaps a misinterpretation of what individual cell trees in this work offer vs. subclone-based approaches in Williams et al, or they are essentially similarly informative.

- In the Abstract the sentence referring to the “orders of magnitude” difference between mutation burden in tumors and healthy tissues should be toned down. It is certainly the case in liver but not necessarily that dramatic in other tissues (authors may want to refer to recent literature by P. Campbell group and colleagues). In any case it deviates the attention from the main purpose of the manuscript that is not on this comparison. This difference in mutation rate is perhaps too much overstated too in pg. 10 given that methods to estimate mutation rate in healthy tissues are so relaxed and heterogeneous. Yet, authors should consider that most reliable method to infer mutation rate in healthy tissues is by fitting data from individuals of different age, and for liver in particular, they can refer to the estimate in Brunner et al (2019) Nature. Here again it is patent that the particular election to normalize mutation rate by cell division is inconvenient as it heavily relies on that estimate, while Brunner et al. present a mutation rate as an independent parameter with dimension site-1 real time-1.

- Fig. 3 is very nice, even if it is limited to some particular true mu:(1-delta) pairs. I would suggest an accompanying figure (e.g. a heatmap), but not necessarily a comprehensive one, showing how uncertainty in parameter estimates changes across a grid search on different true values of mu and (1-delta) extending to other regions of the grid. Do changes in mu affect uncertainty for the same value of delta? It is certainly interesting how uncertainty increases when the death rate approaches the growth rate. The authors point to the fluctuation on the shapes of the trees used for sample generation. I think this deserves a brief reflection. If I am right, I understand that the difficulty in capturing the distribution of bifurcation times with limited leave sized trees in cases where d is similar to b arises from the broad distribution of possible outcomes when fates are balanced in stochastic branching birth-death processes (Bailey, 1964, The elements of stochastic processes with application to the natural sciences). Similar problems arise in healthy tissues (Snippert et al, 2010, Cell; Piedrafita et al, 2020, Nat Commun). Indeed, when commenting on possible reasons for tumors to show elevated values of delta, the underlying tissue architecture and stem cell renewal mode might play important roles too determining the tumor death-to-birth ratios. These factors can be reflected too in the discussion in pg. 11. For instance, the organization of intestinal epithelial stem cells into crypts imposes constraints on clonal expansion between crypts while there is intrinsically elevated competition forces between stem cells belonging to the same crypt (Snippert et al, 2010, Cell).

- The considerations when discussing the cell division rate of the HCC tumor are pertinent but the authors neglect other major factors, such as the significant fraction of the HCC volume occupied by cirrhotic tissue and stroma, or the heterogeneous dynamics between tumor cells depending on their genotype (Darwinian selection) and/or the microenvironment that can greatly affect those estimates. They should tone down the argument that “division rate is realistic, suggesting that the approximation is adequate” – they should do so too when arguing that the fact that the value falls unrealistic when considering random sampling from a large specimen suggests that the sampling ratio may be close to 1.

- I would suggest to change the title for: “Distinguishing excess mutations and increased cell renewal based on tumoral variant allele frequencies”, which is more understandable.

- Regarding the mathematical formulation, the two conditions for m reflected in Eq. 6: is it the intersection between the two sets that is considered? In Eq. 7, better to include a parenthesis for readability, to show that second summation term is nested to the first. Both in Eq. 6 and 7 subindexes i might be pertinent for r_i, m_i, m’_i and m_th,i.

- As a suggestion, variable names m_th, m’, M_obs mut can be explicitly mentioned in the text right before their first use in the respective equations.

- Eq. 7 could benefit from a little explanation in the accompanying text saying that it is a way of weighting the influence of coverage on the discovery of any actual given branch length from just significant mutant read counts, which should thus be < or = to Sum_k l_k. In this sense, I wonder how much this consideration affects the estimates in practice? Can some values for cases where mu is estimated ignoring the significance criterion be represented in Fig. 3? Similarly, in the same paragraph, authors may want to explain that Msig is similar to the total number of accumulated true mutations.

- The definition and extent of a “synthetic dataset” is unprecise. I infer that when “10 synthetic datasets were generated for each true value” it means that 10 query trees were generated for the same given point in the delta grid search, and then “10^4 trees with 10^4 leaves were used for fitting” it means per each dataset?

- Similarly, mu is said to be estimated directly from the data, but since mu_est requires L_sig computation and this depends on T(delta), one infers that it cannot be uncoupled from the delta inference; in other words it is not estimated beforehand but in conjunction with the likelihood for delta, for being a related quantity. Is this correct? This reflection would clarify the procedure.

- What is the criterion when selecting a particular number of simulated trees (10^4)? Is this based on empirical observation, e.g. an optimality criterion like the value for which the error on the estimate gets below a certain threshold or similar? Regarding the choice of 10^4 leaves, it seems to fulfill the number of cells in the empirical HCC dataset, but I wonder if fitting bulk data from a larger tumor sample would require more tree leaves or not necessarily insofar as leaves are representative of the overall subclonal heterogeneity? i.e. Can we assume one branch = one subclone in the formalism used?

- Confidence intervals on MLE values should be provided for the inferred parameters, at least for the empirical dataset. From pg. 14 it is unclear whether a confidence interval is implemented or not but authors could resort to a Likelihood Ratio test or similar in the light of supplementary figures.

- I am just curious to know why not defining delta as a birth-death ratio instead of death-birth ratio. In any case, the election to represent one minus death-to-birth ratio is already sufficiently twisted so that figures might benefit from indicating the extremes corresponding to 0 death and max. death.

- When introducing the likelihood model, it would be useful to state what exactly “observed data D” is. Indeed, is “site frequency spectrum” equivalent to “distribution of variant read counts”? I find less confusing the latter term.

- Authors may acknowledge in pg. 9 if empirical data corresponds with WGS or WES, and whether it involved amplification that could impair the interpretation of the VAF distribution.

TYPOS AND REWORDING

- multiple sites where the term “linage” is used and it should read “lineage”

- (pg. 2) “mutations from individual cells are intermixed” -> rephrase

- (pg. 2) “with the cell linage tree” -> “in terms of a cell lineage tree”

- (pg. 3) “descendance of the extant cells” -> “descendance of any given cell”

- (pg. 9) Perhaps one could simply point to (ii) the level of randomness on tree structure and (iii) the randomness on mutation occurrence as main sources of noise apart from the limited size of trees (i)?

- (pg. 12) “are higher than expected for healthy tissues” -> not true in the case of delta. Rephrase.

- (pg. 14) “from a prescribed distribution” -> an obscure term… A Poisson?

- (pg. 14) “we generate a sample tree” -> “we generate a sample of a given set of simulated trees”

**Have the authors made all data and (if applicable) computational code underlying the findings in their manuscript fully available?**

Reviewer #1: **No: **See review, simulated data missing.

Reviewer #2: **No: **The authors have made (well annotated) code available to generate synthetic data and compute estimates. They could just perhaps include one or two examples in the repository that summarize results in some of the main Fig.

PLOS authors have the option to publish the peer review history of their article (what does this mean?). If published, this will include your full peer review and any attached files.

Reviewer #1: No

Reviewer #2: **Yes: **Gabriel Piedrafita
---

## [Decision Letter · Decision Letter 1]

22 Mar 2022

Dear Dr Szöllősi,

We are pleased to inform you that your manuscript 'Distinguishing excess mutations and increased cell death based on variant allele frequencies' has been provisionally accepted for publication in PLOS Computational Biology.

Best regards,

Teresa M. Przytycka

Associate Editor

PLOS Computational Biology

Douglas Lauffenburger

Deputy Editor

PLOS Computational Biology

Reviewer's Responses to Questions

**Comments to the Authors:**

Reviewer #1: My previous comments have been satisfactorily addressed.

Reviewer #2: The authors have addressed all points raised and have significantly improved the narrative. The details of the model are much clearer and the potential approach limitations/future research directions are well acknowledged in the new version of the Discussion, reason why I consider the current version suitable for publication.

Just two very minor points for consideration (for which no round of revision would be needed by my side):

- In pg. 4 it is explained how cell lineage trees are simulated with continuous-time branch lengths. I guessed but would be nice to state: Is cell turnover simulated as a Poisson process? i.e. cell division rate drawn from an underlying exponential distriibution?

- "Poisson distribution with parameter corresponding to the product" : if "expectancy" parameter is implied, perhaps better specifying so.

Typos:

- pg. 11: "sequnceing"

- pg. 14: "Finnally" and "dyanmics"

**Have the authors made all data and (if applicable) computational code underlying the findings in their manuscript fully available?**

Reviewer #1: Yes

Reviewer #2: None

PLOS authors have the option to publish the peer review history of their article (what does this mean?). If published, this will include your full peer review and any attached files.

Reviewer #1: No

Reviewer #2: **Yes: **Gabriel Piedrafita

---

## [Editor Report · Acceptance letter]

8 Apr 2022

PCOMPBIOL-D-21-01380R1 

Distinguishing excess mutations and increased cell death based on variant allele frequencies

Dear Dr Szöllősi,

I am pleased to inform you that your manuscript has been formally accepted for publication in PLOS Computational Biology. Your manuscript is now with our production department and you will be notified of the publication date in due course.

With kind regards,

Katalin Szabo
